# Effects of Dental Bleaching Agents on the Surface Roughness of Dental Restoration Materials

**DOI:** 10.3390/medicina59061067

**Published:** 2023-06-01

**Authors:** Alexandru Dan Popescu, Mihaela Jana Tuculina, Oana Andreea Diaconu, Lelia Mihaela Gheorghiță, Claudiu Nicolicescu, Cristian Niky Cumpătă, Cristiana Petcu, Jaqueline Abdul-Razzak, Ana Maria Rîcă, Ruxandra Voinea-Georgescu

**Affiliations:** 1Department of Endodontics, Faculty of Dental Medicine, University of Medicine and Pharmacy of Craiova, 200349 Craiova, Romania; alexandrudanpopescu20@gmail.com (A.D.P.); oanamihailescu76@yahoo.com (O.A.D.); leliagheorghita@yahoo.com (L.M.G.); r_ana_maria22@yahoo.com (A.M.R.); 2Department of Engineering and Management of the Technological Systems, Faculty of Mechanics, University of Craiova, 1 Calugareni, 220037 Drobeta-Turnu Severin, Romania; 3Faculty of Dental Medicine, University Titu Maiorescu of Bucharest, 031593 Bucharest, Romania; nikycumpata@yahoo.com (C.N.C.); ruxi0372@yahoo.com (R.V.-G.); 4Department of Prosthodontics, Faculty of Dental Medicine, University of Medicine and Pharmacy of Craiova, 200349 Craiova, Romania; cristiana_croitoru@yahoo.com; 5Department of Infant Care–Pediatrics–Neonatology & Doctoral School, University of Medicine and Pharmacy of Craiova, 200349 Craiova, Romania; jaquelineabdulrazzak90@gmail.com

**Keywords:** dental restoration, microhybrid composite, nanohybrid composite, surface roughness, office bleach, home bleach

## Abstract

*Background and Objectives:* This study aimed to evaluate the surface roughness evolution of several finished and polished composites when bleaching materials are applied. The research was conducted on four microhybrid or nanofilled composites that are used in dental restorations. *Materials and Methods:* For each composite type, 5 samples were selected for control, 5 samples were subjected to the bleaching protocol “office bleach” with 40% hydrogen peroxide, and 5 other samples were subjected to the “home bleach” protocol with 16% carbamide peroxide, resulting in a total number of 60 samples. The surfaces of all the samples were tested for roughness, and the values of the most relevant parameter (Ra), were collected. Comparisons between composites and samples were performed using one-way ANOVA (in Statistical Package for Social Sciences). *Results:* After the bleaching protocol with 40% hydrogen peroxide gel, it was found that the roughness of the group increased considerably compared to the control group, so the highest roughness was found at GC Gradia direct anterior group, and the lowest value was registered for the 3M ESPE Valux Plus group. Following the bleaching protocol with 16% carbamide peroxide (home bleach), it was noted that the sample surfaces were not as affected. In this case, the lowest roughness was found at 3M ESPE Valux Plus group, and the highest roughness was registered for the GC G-aenial anterior group. Following the interpretation of the results, all four types of dental composites tested showed significant surface roughness differences between the groups subjected to bleaching protocols and those kept as control (*p* < 0.05). *Conclusions:* The surfaces of the samples were affected by the bleaching protocols by increasing the roughness compared to the control samples.

## 1. Introduction

Composite materials are among the most widely used dental restoration materials. Technologies and materials are advancing rapidly, and at present, composite materials with different properties and particle sizes are used quite frequently in the filling. By combining the aesthetic properties of composite materials with microfilling and the mechanical properties of hybrid composites, nanohybrid composite materials appeared [1,2].

Resin composite’s three basic components—an inorganic filler, an organic polymeric matrix, and a silane coupling agent—make up their fundamental structure. The composition and microstructure of resin composites have been proven to affect both their mechanical characteristics and aesthetic appeal [3,4,5].

Some studies have shown that there are no significant differences between the clinical performances of the two types of composites (nanohybrid and microhybrid) [6].

However, there are also studies mentioning that microhybrid composites do not have clinical success over time, especially in terms of maintaining the original color [7].

In preserving the aesthetics of these resins, an important role is played by finishing and polishing the dental restoration. If proper finishing and polishing are not performed, roughness at the surface of the composites can influence the adhesion but also the retention of bacterial plaque, which can lead to an increase in the risk of periodontal disease and dental caries [8,9,10].

Restorations with a smooth surface are thus obtained, reducing the risk of fracture, preventing color changes in the composite and improving the patient’s comfort at the same time [10,11,12].

There is currently insufficient evidence to support the superiority of nanofilled or submicron materials in terms of surface smoothness and gloss, according to a systematic review of in vitro studies comparing the surface characteristics of composites with nano- or submicron-sized fillers and conventional composites [5,6,7].

In order to improve the aesthetic appearance of patients, doctors very frequently resort to dental bleaching, carried out with professional products [13,14,15].

For dental bleaching, the most used substances are represented by hydrogen peroxide and carbamide peroxide of different concentrations depending on the bleaching technique prescribed by the dentist. In the case of the bleaching technique in the dental office, 35% up to 40% hydrogen peroxide preparations are used [16,17].

Bleaching techniques performed at home by patients but prescribed by dentists most commonly use carbamide peroxide in the form of a gel of various concentrations, from 10% to 15% or even 16% [18].

Tooth-bleaching products also come into contact with the surface of dental restorations, affecting their mechanical, physical and aesthetic properties [19,20,21,22].

The surface quality of a sample can be quantified by measuring roughness, which shows the fissures, streaks or traces resulting from a particular process of working or finishing/polishing. The profile of these traces can be described using the following parameters: Rz (the average roughness of the surface), Ra (the average arithmetic deviation of the profile) and Rq (the average quadratic deviation of the heights of the profile roughness) [23,24,25].

The purpose of this study was to evaluate the effects of a professional bleaching product and a home bleaching product on the surface morphology of four different composite materials, three microhybrid (Gradia Direct Anterior, G-aenial Anterior, Valux Plus) and one nanohybrid (Filtek Z550).

The null hypothesis (h0) was that the use of tooth bleaching agents had no effect on the surface roughness of dental restorative composite materials.

This study is significant because it will add to the literature on the mechanical changes undergone by composite materials under the action of tooth-bleaching substances and will provide clinicians with evidence-based recommendations. Ultimately, this can lead to improved clinical outcomes and a higher success rate for teeth bleaching treatment.

## 2. Materials and Methods

For the study, the following composite materials, as presented in Table 1, were selected.

Test samples from each composite presented in Table 1 were selected, properly labeled and documented. The samples were made using a stainless-steel device, which produced composite discs that are 10 ± 0.1 mm in diameter and 2.5 ± 0.1 mm thickness (maximum layer thickness approved by each composite manufacturer for shade A2). In order to make the samples, the composite material was applied to the device. The samples were prepared by compressing two glass plates to ensure uniformity and to eliminate air bubbles, using finger pressure to remove excess resin and ensure a flat surface. Photopolymerization was carried out successively, 20 s above the higher placed plate and 20 s below the lower located plate (in total 40 s), using the led E. Woodpecker wireless lamp with a light intensity of 700 mW/cm^2^ for photopolymerization. Because glass plates were used, increasing the distance from the composite material, bidirectional photopolymerization was carried out so that the process could be correct and complete.

An experimental protocol with clinical relevance was adopted following the manufacturer’s instructions. The photopolymerization time was in line with the specifications of each manufacturer.

After photopolymerization, the samples were kept in distilled water at a temperature of 37 °C for 24 h, after which they were subjected to the finishing and polishing process with the help of the Super Snap system. The finishing and polishing procedure for all samples was carried out by a single dentist at conventional RPMs, following the manufacturer’s instructions. After this process, the samples were measured using a micrometer, and the final thickness of the samples was 2.4 ± 0.1 mm.

Protocol was identical for all four types of composite materials: G-aenial anterior, Gradia direct anterior, Filtek Z550 and Valux Plus. For each selected material, 15 samples were made (5 control, 5 subjected to the “office bleach” protocol and 5 subjected to the “home bleach” protocol), resulting in a total number of 60 samples.

The control samples were represented by composite discs. The second type consisted of discs made of the same material as the control type on which the dental bleaching protocol “office bleach” was applied with 40% hydrogen peroxide gel, in a layer of 1 mm, by two applications of 20 min each in the same session. After each application, the samples were washed with a saline solution of NaCl 0.9%. Finally, they were stored in containers with a saline solution of NaCl 0.9% at the temperature of 37 °C. The last type of samples used discs of the same composite materials as the control type, on which the dental bleaching protocol “home bleach” of the sample was applied, with 16% carbamide peroxide gel, in a layer of 1 mm, 6 h a day for 7 days. Between the bleaching sessions, the samples were held in a saline solution of NaCl 0.9% at 37 °C.

In order to be tested and to avoid the appearance of cracks upon contact with the vice of the equipment, the samples were cold-mounted into a special resin, as shown in Figure 1.

The roughness of the samples was determined using a Surtronic 25 profilometer, Taylor Hobson, with the following characteristics: calibration limit—300 μm, resolution—0.01 μm, examination length—0.25–25 mm, examination speed—1 mm/sec, and computer connection RS232.

The measurement distance (8 mm) was set with the help of the dedicated TalyProfile Silver software, and the data acquisition was made (the results were saved in the form of text files but also in graphic form), namely, the roughness profile and the roughness parameters according to STANDARD ISO 4287-2003 replaced by SR EN ISO 21920-2:2022. The procedure was repeated for each sample of each type of composite.

According to STANDARD ISO 4287-2003, replaced by SR EN ISO 21920-2:2022, the roughness parameters are described in Table 2.

The measured roughness parameters were initially centralized using Microsoft Excel (San Francisco, CA, USA). The parameter Ra was chosen to define the overall roughness of each sample included in this study and was used for the subsequent statistical analysis. The present study was related to the Ra parameter because it is the most used roughness parameter and represents the average arithmetic deviation of roughness according to STANDARD ISO 4287-2003, replaced by SR EN ISO 21920-2:2022.

Statistical tests were performed using Statistical Package for Social Sciences (SPSS), version 20 (IBM Corp., Foster City, CA, USA) based on the primary data acquired before. The descriptive analysis reflects continuous variables expressed as mean ± standard deviation unless stated otherwise. Statistical analysis was based on Shapiro–Wilk’s test for data normality analysis, Levene’s test of equality of variances, and one-way ANOVA for group comparisons. For this study, the value *p* < 0.05 was considered statistically significant.

## 3. Results

The profiles were saved as .txt files, and the charts were plotted in the Origin software. Charts were made for each type of composite following the three conditions, namely control, “office bleach” and “home bleach”.

Table 3 summarizes the mean values of the most important parameters of roughness computed for each group.

The evolution of roughness parameters Ra, Rz and Rq are presented in Figure 2, and in the case of control groups, according to the roughness parameter Ra, the lowest value was registered at 3M ESPE Valux Plus control group, and the highest value was found at GC Gradia direct anterior control group.

After the bleaching protocol with 40% hydrogen peroxide (office bleach), it was found that the roughness of the groups increased considerably compared to the control groups, so the highest roughness was found at GC Gradia direct anterior office bleach group (Ra = 0.915 μm), and the lowest value was registered for the 3M ESPE Valux Plus office bleach group (Ra = 0.311 μm) according to Figure 2.

Following the bleaching protocol with 16% carbamide peroxide (home bleach), it was noted that the groups’ surfaces were not so affected as in the case of the surfaces of the groups bleached with 40% hydrogen peroxide. In the case of these groups, the lowest roughness was found at 3M ESPE Valux Plus home bleach group (Ra = 0.220 μm), and the highest roughness value was registered for the GC G-aenial anterior home bleach group (Ra = 0.622 μm), as is showed in Figure 2 and Table 3.

### 3.1. GC G-Aenial Anterior Composite—Analysis

As can be seen from Figure 3, the roughness profile of the GC G-aenial anterior control group decreases below the midline after approximately 4.5 mm. This is due to the fact that the measured surface is not perfectly flat, as it shows certain bumps.

Given that the most important roughness parameter values are shown in Figure 4a–c and Table 3. Ra (0.249 μm) and Rq (0.324 μm) have subunit values, and the roughness profile differs greatly from a profile of a perfectly flat surface. Moreover, the peaks and gaps, respectively, in roughness, are almost unnoticeable. The values of the other roughness parameters are consistent with the value of the arithmetic mean deviation of the roughness Ra.

The roughness profile of the GC G-aenial anterior office bleach group, according to Figure 3, is not a linear one and consists of 3 zones, as follows: the first zone, up to about 0.6 mm, is below the midline; the second zone, up to 5.2 mm, is found above the midline; and the third zone, from 5.2 mm to 8 mm, is below the midline. In the third zone, at a distance of about 6.5 mm, three more prominent peaks appear, which confirms the increase in the roughness value of Ra (0.565 μm) as registered in Figure 4b compared to the GC G-aenial anterior control group (Ra = 0.249 μm) as seen in Figure 4a. This aspect is also justified by the higher value of the total profile height—Rt =12.4 μm compared to Rt = 3.55 μm for the GC G-aenial anterior control group.

The roughness profile of the GC G-aenial anterior home bleach group presented in Figure 3 shows two zones, one up to a distance of 2.8 mm above the midline and the other below the midline. In the second zone at distances of approximately 5.3 mm and 6 mm, respectively, two larger peaks are present compared to those present in the case of GC G-aenial anterior home bleach group, and this aspect is justified by a higher roughness value Ra = 0.622 μm presented in Figure 4c compared to the roughness value of GC G-aenial anterior office bleach group. At the same time, the value of the parameter Rt = 16.3 μm is higher than the values Rt for GC G-aenial anterior control group and GC G-aenial anterior office bleach group.

For the two bleached groups, it was found that higher roughness values were obtained compared to the roughness of the control groups, according to Table 3.

A one-way ANOVA was conducted to determine if the roughness parameter (Ra value) was different for the three groups of samples identified for this type of composite. There were no outliers, as assessed by a boxplot; data were normally distributed for each group, as assessed by Shapiro–Wilk test (*p* > 0.05); additionally, there was homogeneity of variances, as assessed by Levene’s test of homogeneity of variances (*p* = 0.717). The roughness was statistically significantly different for the three groups, F(2, 12) = 11,480.306, *p* < 0.0005. The roughness value increased from the control group (0.248 ± 0.0038) to the office bleach group (0.564 ± 0.0038) and the home bleach group (0.622 ± 0.0048), in that order. Tukey post hoc analysis revealed that the increase from control to office bleach (0.316, 95% CI (0.308 to 0.323)) was statistically significant (*p* < 0.0005), as well as the increase from control to home bleach (0.373, 95% CI (0.366 to 0.380), *p* < 0.0005), and the increase from office bleach to home bleach (0.057, 95% CI (0.050 to 0.064), *p* < 0.0005) (Table 4).

### 3.2. GC G-Gradia Anterior Composite—Analysis

The roughness profile of the GC Gradia direct anterior control, according to Figure 5, is found with one area up to a distance of about 4.8 mm below the midline and with the other area above; this aspect is justified by the fact that the surface is not perfectly flat.

According to Figure 6a, the value of the average arithmetic deviation of roughness Ra is 0.365 μm in group control.

The roughness profile of the GC Gradia direct anterior office bleach group shows more areas compared to the profile of the GC Gradia direct anterior control group, according to Figure 5, which shows that the surface of the bleached group has several bumps. Regarding the value of roughness Ra, it is higher than that of the control group, i.e., 0.915 μm. The values of the other parameters are found in Figure 6b.

The roughness value according to Figure 6c of Ra from the GC Gradia direct anterior home bleach group is higher than that of the control group but lower than that of the GC Gradia direct anterior office bleach group, as are the other roughness parameters consistent with the value of Ra presented in Table 3.

A one-way ANOVA was conducted to determine if the roughness parameter (Ra value) was different for the three groups of samples identified for this type of composite. There were no outliers, as assessed by a boxplot; data were normally distributed for each group, as assessed by Shapiro–Wilk test (*p* > 0.05); additionally, there was homogeneity of variances, as assessed by Levene’s test of homogeneity of variances (*p* = 0.174). The roughness was statistically significantly different for the three groups, F(2, 12) = 32,632.930, *p* < 0.0005. The roughness (Ra score) increased from the control group (0.364 ± 0.0049) to home bleach (0.371 ± 0.0024) and office bleach (0.914 ± 0.0039), in that order. Tukey post hoc analysis revealed that the increase from control to office bleach (0.550, 95% CI (0.543 to 0.556)) was statistically significant (*p* < 0.0005), as well as the increase from control to home bleach (0.007, 95% CI (0.0004 to 0.013), *p* = 0.037), and the increase from home bleach to office bleach (0.543, 95% CI (0.536 to 0.549), *p* < 0.0005) (Table 4).

### 3.3. 3M ESPE Filtek Z550 Composite—Analysis

The roughness profile of the 3M ESPE Filtek Z550 control group shows three zones indicating the bumps that exist on the surface of the area, according to Figure 7.

The roughness value of Ra is 0.175 μm, as is presented in Figure 8a, and the other roughness parameters are consistent with this value.

In the case of the 3M ESPE Filtek Z550 office bleach group, at a distance of about 1.5 mm, the presence of a peak is observed in Figure 7 and, due to this, the value of the parameter Rt is 20.3 μm, which is higher than the values registered for groups 1 to 5. The roughness value of Ra (0.487 μm), as registered in Figure 8b, is higher than the roughness of the control group.

3M ESPE Filtek Z550 home bleach group shows a roughness (Ra = 0.232 μm) as registered in Figure 8c, higher than 3M ESPE Filtek Z550 control group but lower than 3M ESPE Filtek Z550 office bleach group.

A one-way ANOVA was conducted to determine if the roughness parameter (Ra value) was different for the three groups of samples identified for this type of composite. There was one outlier, as assessed by a boxplot, which was kept in the analysis (the one-way ANOVA performed with this outlier replaced by a mean value yielded similar results); data were normally distributed for each group, as assessed by Shapiro–Wilk test (*p* > 0.05); and there was homogeneity of variances, as assessed by Levene’s test of homogeneity of variances (*p* = 0.299). The roughness was statistically significantly different for the three groups, F(2, 12) = 3982.400, *p* < 0.0005. The roughness (Ra score) increased from the control group (0.175 ± 0.0030) to home bleach (0.232 ± 0.0047) and office bleach (0.486 ± 0.0084), in that order. Tukey post hoc analysis revealed that the increase from control to office bleach (0.311, 95% CI (0.301 to 0.321)) was statistically significant (*p* < 0.0005), as well as the increase from control to home bleach (0.057, 95% CI (0.047 to 0.066), *p* < 0.0005), and the increase from home bleach to office bleach (0.254, 95% CI (0.244 to 0.264), *p* < 0.0005) (Table 4).

### 3.4. 3M ESPE Valux plus Composite—Analysis

After bleaching, the roughness profile of the 3M ESPE Valux Plus office bleach group, according to Figure 9, is no longer linear, and those bumps occur.

For the 3M ESPE Valux Plus control group, the lowest roughness value was registered, i.e., Ra = 0.138 μm according to Figure 10a and Table 3.

The roughness value (Ra = 0.311μm) registered in Figure 10b is higher than the value of the 3M ESPE Valux Plus control group.

The roughness profile of the 3M ESPE Valux Plus home bleach group is approximately linear, according to Figure 9; the roughness value Ra = 0.22 μm, as registered in Figure 10c, is higher than that of the 3M ESPE Valux Plus control group but lower than the 3M ESPE Valux Plus office bleach group according to Figure 10b and Table 3.

A one-way ANOVA was conducted to determine if the roughness parameter (Ra value) was different for the three groups of samples identified for this type of composite. There were no outliers, as assessed by a boxplot; data were normally distributed for each group, as assessed by Shapiro–Wilk test (*p* > 0.05); and there was homogeneity of variances, as assessed by Levene’s test of homogeneity of variances (*p* = 0.372). The roughness was statistically significantly different for the three groups, F(2, 12) = 2745.390, *p* < 0.0005. The roughness (Ra score) increased from the control group (0.137 ± 0.0042) to home bleach (0.219 ± 0.0041) and office bleach (0.311 ± 0.0024), in that order. Tukey post hoc analysis revealed that the increase from control to office bleach (0.173, 95% CI (0.167 to 0.180)) was statistically significant (*p* < 0.0005), as well as the increase from control to home bleach (0.082, 95% CI (0.075 to 0.088), *p* < 0.0005), and the increase from home bleach to office bleach (0.091, 95% CI (0.085 to 0.097), *p* < 0.0005) (Table 4).

For each type of composite, the variation in roughness was computed as a difference between the initial roughness measured before the use of the bleaching agent and the roughness measured afterward, both for office bleaching and home bleaching.

For home bleaching, a one-way ANOVA was conducted to determine if the variation in roughness (Ra variation) was different for the four types of composites: GC G-aenial anterior, GC Gradia direct anterior, 3M ESPE Filtek Z550 and 3M ESPE Valux Plus, each one with five discs each. There were no outliers, as assessed by boxplot; data were normally distributed for each group, as assessed by Shapiro–Wilk test (*p* > 0.05); and there was homogeneity of variances, as assessed by Levene’s test of homogeneity of variances (*p* = 0.239). Roughness variation was statistically significantly different between the four types of composites, F(3, 16) = 7986.519, *p* < 0.0005. Ra variation is indicated in Table 5. Tukey post hoc analysis revealed that the group differences were statistically significant, *p* < 0.0005.

For office bleaching, a similar one-way ANOVA was conducted to determine if the variation in roughness (Ra variation) was different for the four types of composites. There was one outlier, as assessed by a boxplot, which was kept in the analysis (the one-way ANOVA performed with this outlier replaced by a mean value yielded similar results); data were normally distributed for each group, as assessed by Shapiro–Wilk test (*p* > 0.05); and there was homogeneity of variances, as assessed by Levene’s test of homogeneity of variances (*p* = 0.237). Roughness variation was statistically significantly different between the four types of composites, F(3, 16) = 4513.880, *p* < 0.0005. Ra variation is indicated in Table 5. Tukey post hoc analysis revealed that there were no significant differences between the Ra variation between groups GC G-aenial anterior and 3M ESPE Filtek Z550 (*p* = 0.483), but all other group differences were statistically significant, *p* < 0.0005.

## 4. Discussion

The null hypothesis was that the use of bleaching chemicals had no impact on the surface roughness of dental restorative composite materials. The null hypothesis is rejected by the findings of the preceding investigation. These observed changes in the surface roughness of the composites have clinical significance.

The bleaching procedure is a topic of extensive discussion in modern dentistry because it is a technique that is quite easy to achieve and, at the same time, conservative, which has been shown to be effective in improving the color of natural teeth, but which has generated many contradictory discussions regarding its effects on dental restoration materials and especially on composites. In current practice, peroxides are used in gel form in different concentrations from 4% to 22% in home bleaching techniques and in concentrations of 25% up to 40% concentrations for bleaching techniques in the dental office [26,27,28,29,30,31,32].

For patients, bleaching treatment is a non-invasive therapeutic procedure that does not pose problems; however, some authors noticed adverse effects of this treatment, not only at the level of soft tissues in the oral cavity but also in the hard dental tissues. The same authors concluded that the adverse effects of bleaching agents also occur at the level of already existing dental restoration materials. The harmful effects are closely related to the concentration of bleaching agents, the time of contact with dental hard tissues/dental restoration materials, and the acidity of the gels used [33].

The emergence of new materials used in dental restorations with a color similar to teeth, new bleaching materials and techniques, and the increasing demands of patients have led to the development and study of various types of composite materials. The types of microhybrid composites have been successfully used both in the restoration of the teeth in the anterior and posterior areas, and the nanohybrid composites that have appeared in recent years have had good clinical success and an attractive appearance. Of course, both the physical and mechanical properties of dental restoration materials, such as composite materials, contribute to the success of tooth restoration. These properties are durability; abrasion resistance; restoration appearance; surface smoothness; roughness; and, of course, in the end, patients’ satisfaction [34,35,36].

Dental restorations using microhybrid or nanohybrid composites provide a special aesthetic. However, it is very important for dentists to take into account the functionality, stability and structural integrity of these materials used in the restoration of teeth. Nanofilling composites have a higher translucency than those with microfilling and retain their physico-chemical properties better [37,38,39,40].

Regarding the working procedure, in the present study, the samples were made according to the instructions given by the manufacturer and the data found in the specialized literature, namely studies made on the same types of composites or on similar ones. Moreover, increasing the photopolymerization time of the composites did not have negative effects and even improved the mechanical properties of the composites [41,42].

Roughness on the surface of the materials used in dental restorations influences both adhesion and retention of bacterial plaque. This eventually leads to an increased risk of tooth decay and other complications in the oral cavity [13,43].

Roughness on the surface of composites can also influence the color of dental restoration. Due to the increase in adhesion and retention of chromogenic food debris, the susceptibility of staining of the respective composites also increases, the gloss being subsequently affected. Other important elements that can influence the roughness of composites are represented by the dexterity of the dentist, the technique and system used during finishing and polishing, and the patient’s diet and hygiene; all these factors influence the roughness of the composites [44,45].

The results of the current study suggest that the roughness increased at the surfaces of all samples subjected to the bleaching protocols, in complete accordance with specialty literature [46,47].

Other studies reported increases in the roughness of the surfaces of the resinous materials used, but also the appearance of cracks, pores and even scratches. The samples were analyzed using the electron microscope after undergoing the bleaching protocol using carbamide peroxide [48,49,50].

To combat this increase in roughness, some authors recommend polishing the dental obturations restored with composite materials after undergoing the bleaching protocol. A recent study showed that bacterial adhesion in gyomers is lower compared to other types of composite materials, although roughness values were similar [51].

During the bleaching protocol, a release of free radicals from bleaching agents that can penetrate into the composite material and also break the filler-resin bond may occur. This alteration of the bond leads to an increase in roughness and implicitly may cause an increase in the adhesion of microorganisms with the cariogenic potential to the surface of composites subjected to the bleaching protocol [52].

Recent studies have been carried out on both traditional composites and bulk-type composites (with high viscosity as well as with low viscosity). These composites have undergone bleaching protocols with both 40% hydrogen peroxide and 10% carbamide peroxide. Following the testing carried out, it was concluded that the Ra (roughness) of the surfaces was higher after the bleaching procedures. Bulk-type composites with high viscosity recorded a higher roughness than conventional nanofilled and low-viscosity ones [53,54,55,56].

As a result of the increase in Ra after the bleaching procedures, irregularities were observed at the sample surfaces and the appearance of small grooves at the morphology of the sample surface. These irregularities can occur due to structural changes in composite materials subject to bleaching. Both the free radicals released by peroxides and the absorption of water during the bleaching protocol can cause both increased roughness and degradation of the surface of composite materials. This absorption process can cause the matrix of composites to weaken and subsequently degrade the surface of the respective resins with the exposure of the inorganic load and the increase in roughness [57,58,59,60].

Yap et al. concluded that maintaining the bleaching agent for a long time through the “home bleach” technique on the surfaces of restoration materials such as composites can cause an increase in their roughness. In addition to this, prolonged application of the bleaching agent can increase the roughness and the appearance of microcracks on the surface of composites. Some researchers claim that these changes can occur in composite surfaces as a result of the interaction between bleaching agents and the composite matrix. Others support that the changes in the surface of dental restoration materials such as composites are due to the difference in the composition of the matrix of each composite [61,62,63,64,65,66,67].

In 2008, Dogan et al. conducted a similar study, determining the roughness of some composite materials. Following testing, the researchers showed that the roughness of the surfaces of the bleached samples decreased, contrary to the results obtained in this study [68].

In order to highlight the importance of the time of photopolymerization of composite materials better, Özduman et al. made three different photopolymerization times (10, 20 and 30 s) for two types of different composites that were subsequently bleached. During the testing, at the surface of those samples, the roughness increased following the application of the bleaching protocol. The authors concluded that there was no such significant increase in roughness when observing the photopolymerization times indicated by the manufacturers [69,70].

In 2018, the bleaching and testing of the surface roughness of three different types of composite materials were carried out, namely: Estelite^®^ α, FiltekTM P90 and FiltekTMZ250 XT. The samples were bleached with the help of carbamide peroxide in the form of a gel of 15% concentration for 8 h a day over a period of 14 consecutive days. After testing the roughness, a more intense attack of carbamide peroxide on FiltekTM Z250 XT was observed compared to the other two composites. The authors concluded that the result might be due to the decomposition of carbamide peroxide in urea and hydrogen peroxide in aqueous solution. Then the hydrogen peroxide enters the surface of the restorative material [71].

Rodrigues CS. et al. bleached each microhybrid and nanofilled composite sample with 35% hydrogen peroxide. The samples were subjected to the bleaching procedure of 2 bleaching sessions per day for 20 min, after which the samples were placed in containers with red wine for 15 min daily throughout the experiment. The authors wanted to highlight the importance of polishing samples of bleached composites and then subjecting them to the process of staining with red wine. It was concluded that the polishing of the composite samples after the bleaching protocol improved the smoothness of the sample surfaces and the color stability of the respective composites. The samples that were not polished after the bleaching underwent greater changes in smoothness and color after the staining process (aging), with the coloring tendency being higher in nanofilled composites [72].

The link between the increase in roughness and the formation of bacterial biofilm has been demonstrated, with nanofilled composites and resin-modified glassionomer cement being used in the testing process. For the bleaching of the samples, 10% carbamide peroxide and 40% hydrogen peroxide were used, and then the process of biofilm formation on them was observed. Roughness increased at the surfaces of all samples subjected to the bleaching protocols, and the increase in the biofilm of Streptococcus mutans and Streptococcus sanguinis was observed both at the nanofilled composites and at the resin-modified glassionomer cement. Streptococcus mutans is a cariogenic pathogen that, when introduced into an acidic environment, becomes much more prevalent in the biofilm. Thus, it increases the risk of caries initiation as well as its progression [73].

The study’s novelty stems not only from the fact that a greater number of surface roughness parameters of the latest generation composites were determined but also from the fact that bleaching materials were applied at a higher concentration, making a comparison with previously published works difficult. This study informs clinicians about the level of risk associated with sustaining a composite restoration with increased surface roughness and the potential for plaque retention.

Regarding the clinical significance of the study, the findings imply that the alteration of the surface roughness of composites following the application of some whitening treatments necessitates the repolishing of the restorations or the placement of fresh composite fillings.

### 4.1. Limitations of the Study

The limitations of this study should be carefully considered when interpreting the results. First, the study was conducted in vitro, which means the results may not accurately reflect what happens in the clinical setting. In vitro studies are usually performed in controlled environments, where variables that could affect the results are minimized.

Therefore, further studies are needed to determine the influence of these bleaching agents on a wider range of composite materials.

Summarily, while this study provides valuable information on surface roughness changes in fillings, its limitations suggest that further research is needed to determine the broader applicability and safety of these findings.

### 4.2. Recommendations for Further Research

There is a need for continued investigation to build a further understanding of the efficacy and safety of teeth bleaching. The implementation of these recommendations will help develop improved tooth bleaching protocols and ultimately improve the clinical outcomes of bleaching treatment.

## 5. Conclusions

Bleaching agents containing 40% hydrogen peroxide and 16% carbamide peroxide increase the surface roughness of composite materials. For the two types of composite materials studied, the highest surface roughness was obtained after the action of 40% hydrogen peroxide.

The microhybrid composite recorded higher surface roughness values compared to the nanohybrid composite after the action of bleaching agents.

However, caution is warranted in the interpretation of findings, as further research is needed to confirm the results on larger-scale and in vivo studies. Specifically, future studies should consider expanding the sample size and investigating more types of physiognomic dental restorative materials. Ultimately, a better understanding of the optimal concentration and duration of application for these filling materials is needed to improve the clinical management of physiognomic fillings.

## Figures and Tables

**Figure 1 medicina-59-01067-f001:**
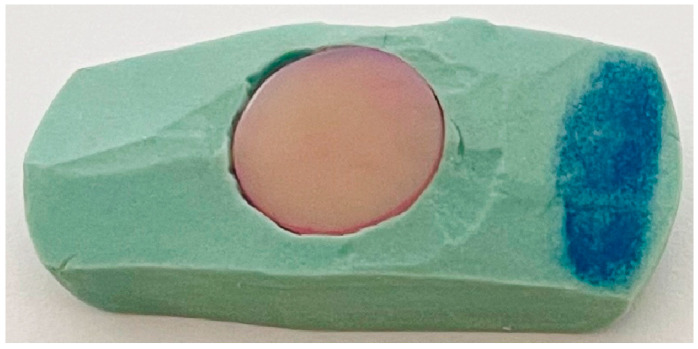
Mounted sample.

**Figure 2 medicina-59-01067-f002:**
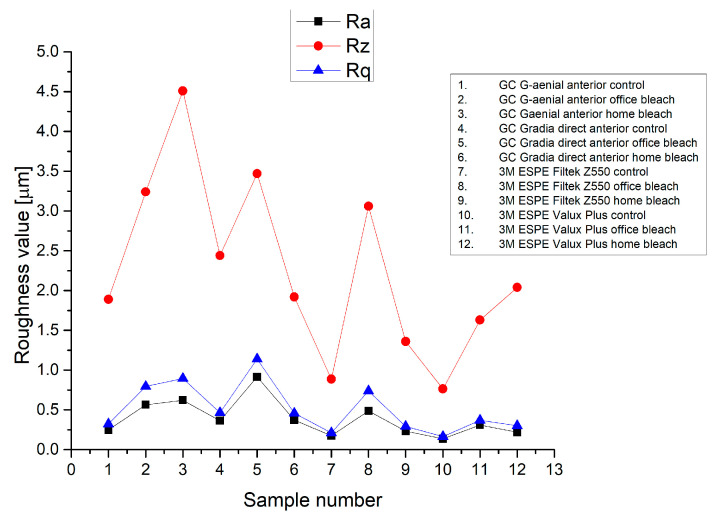
Evolution of roughness parameters Ra, Rz and Rq.

**Figure 3 medicina-59-01067-f003:**
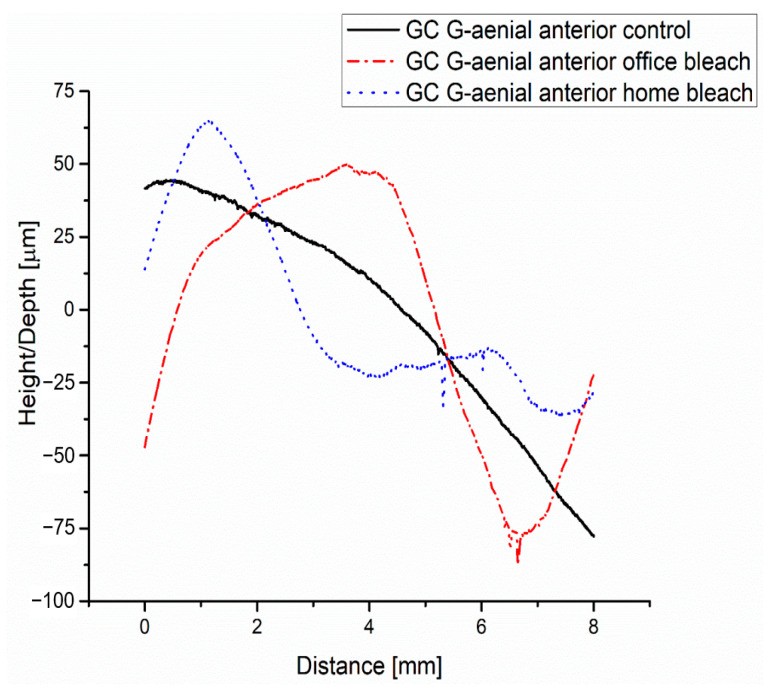
Roughness of the GC G-aenial anterior groups (control, office bleach and home bleach): profiles.

**Figure 4 medicina-59-01067-f004:**
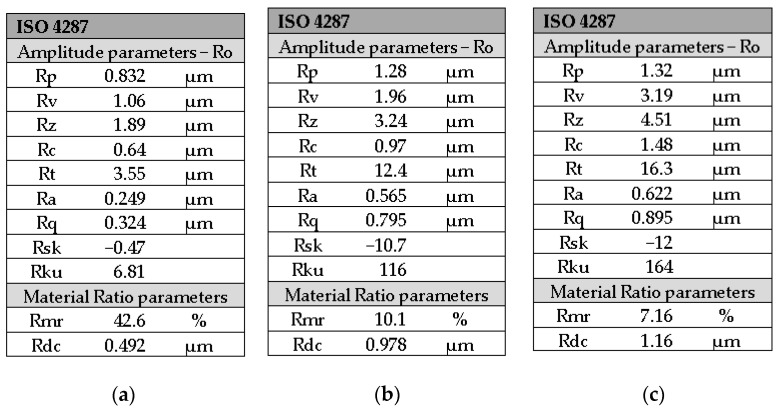
Roughness parameters of the GC G-aenial anterior for groups: (**a**) control; (**b**) office bleach; (**c**) home bleach.

**Figure 5 medicina-59-01067-f005:**
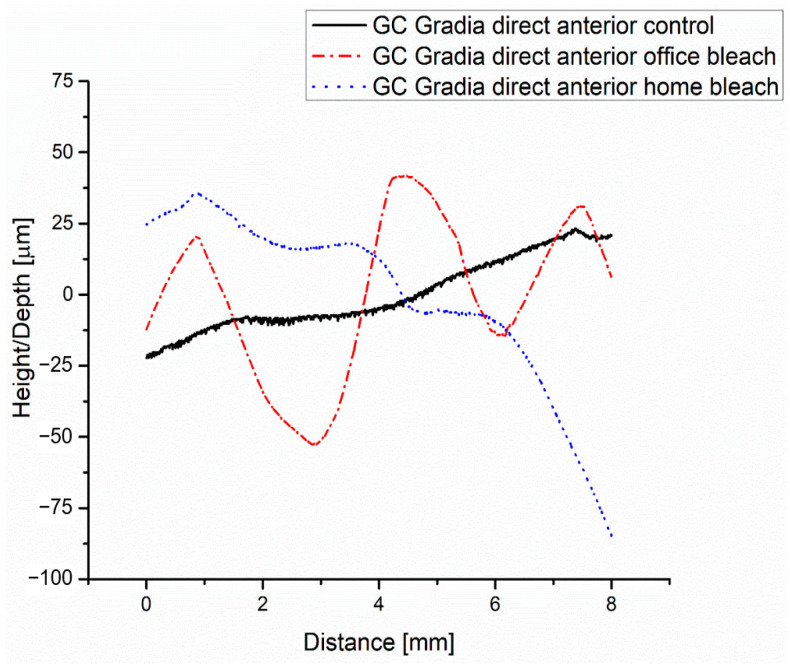
Roughness of the GC Gradia direct anterior groups (control, office bleach and home bleach): profiles.

**Figure 6 medicina-59-01067-f006:**
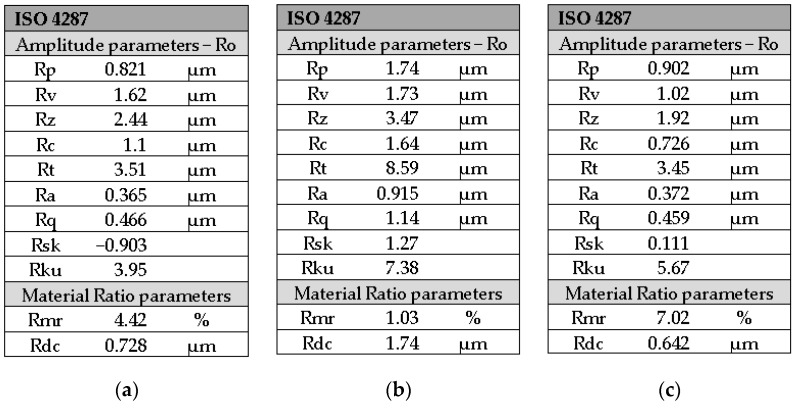
Roughness parameters of the GC Gradia direct anterior for groups: (**a**) control; (**b**) office bleach; (**c**) home bleach.

**Figure 7 medicina-59-01067-f007:**
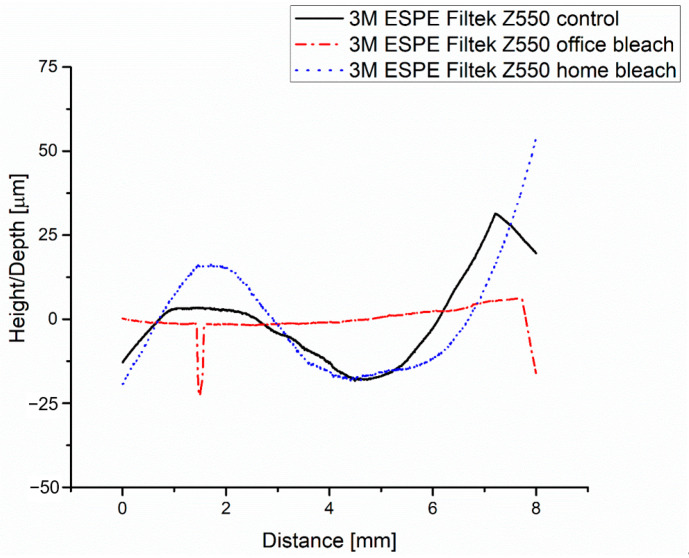
Roughness of the 3M ESPE Filtek Z550 groups (control, office bleach and home bleach): profiles.

**Figure 8 medicina-59-01067-f008:**
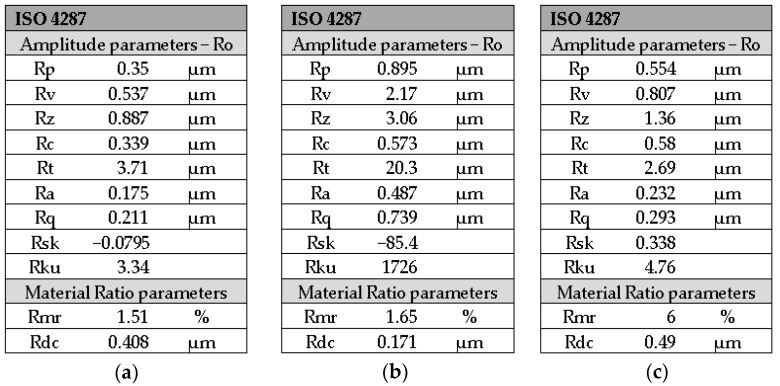
Roughness parameters of the 3M ESPE Filtek Z550 for groups: (**a**) control; (**b**) office bleach; (**c**) home bleach.

**Figure 9 medicina-59-01067-f009:**
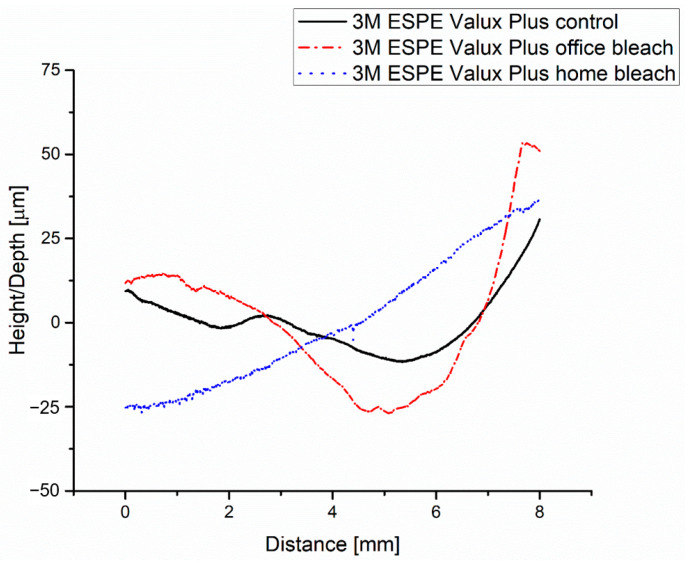
Roughness of the 3M ESPE Valux Plus groups (control, office bleach and home bleach): profiles.

**Figure 10 medicina-59-01067-f010:**
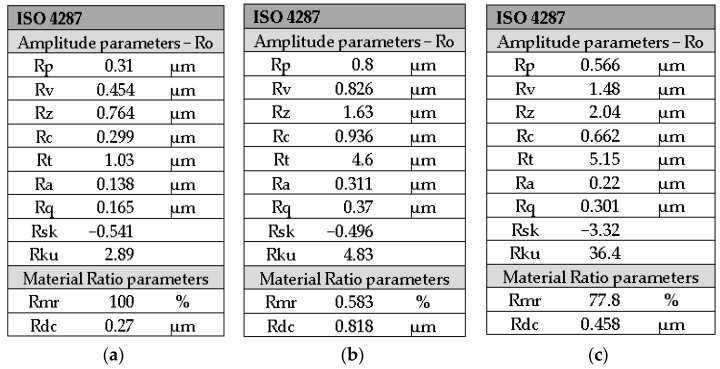
Roughness parameters of the 3M ESPE Valux Plus for groups: (**a**) control; (**b**) office bleach; (**c**) home bleach.

**Table 1 medicina-59-01067-t001:** Composite materials.

Name of Composite	Type of Composite	Manufacturer	Composition
Gradia Direct Anterior	Microhybrid composite	GC Tokyo, Japan	Microhybrid composite material matrix: UDMA, dimethacrylates, trimethacrylates. Fillers: silica and prepolymerised resin fillers (73 wt%).
G-aenial Anterior	Microhybrid composite	GC Tokyo, Japan	Microhybrid composite matrix: UDMA, dimethacrylate comonomers. Fillers: prepolymerised silica and strontium fluoride-containing fillers (76 wt%).
Filtek Z550	Nanohybrid composite	3M ESPE, Minnesota USA	Nanohybrid composite matrix: BIS-GMA, TEGDMA, UDMA, BIS-EMA, PEGDMA. Fillers: silica-zirconia fillers, non-agglomerated/non-aggregated silica particles, filler loading (82 wt%).
Valux Plus	Microhybrid composite	3M ESPE, Minnesota USA	Microhybrid composite matrix: BIS-GMA, TEGDMA. Fillers: zirconia/silica loading is 66% by volume with a particle size range of 3.5 to 0.01 micron.

UDMA: Urethane Dimethacrylate. BIS-GMA: Bisphenol A-Glycidyl Methacrylate. TEGDMA: Tri-ethylene Glycol Dimethacrylate. BIS-EMA: Bisphenol A-Ethoxylated Dimethacrylate. PEGDMA: Polyethylene Glycol Dimethacrylate.

**Table 2 medicina-59-01067-t002:** Roughness parameters.

Roughness Parameters	Signification
Rp	The upper area, between the middle line and the line parallel to it, tangent to the highest peak.
Rv	The lower area, located between the middle line and the line parallel to it passing through the deepest gap of the roughness profile.
Rz	The maximum height of the irregularity of the profile in ten points.
Rc	Average height of profile elements: defined on the assessment length.
Rt	Total height of the profile: the height between the deepest valley and the highest peak on the evaluation length.
Ra	Average arithmetic deviation of roughness.
Rq	The average quadratic deviation of the heights of the profile roughness.
Rsk	Asymmetry of the assessed profile: the asymmetry of the height distribution, defined on the sampling length.
Rku	Trial of the assessed profile: the clarity/accuracy of the height distribution, defined on the sampling length.

**Table 3 medicina-59-01067-t003:** Values of roughness parameters (mean ± SD).

Number	Group Name	Rz [µm]	Ra [µm]	Rq [µm]
1	GC G-aenial anterior control	1.888 ± 0.0455	0.249 ± 0.0038	0.324 ± 0.0028
2	GC G-aenial anterior office bleach	3.242 ± 0.0239	0.565 ± 0.0038	0.795 ± 0.0048
3	GC G-aenial anterior home bleach	4.508 ± 0.0487	0.622 ± 0.0048	0.895 ± 0.0095
4	GC Gradia direct anterior control	2.438 ± 0.0449	0.365 ± 0.0049	0.466 ± 0.0062
5	GC Gradia direct anterior office bleach	3.47 ± 0.0485	0.915 ± 0.004	1.144 ± 0.0493
6	GC Gradia direct anterior home bleach	1.918 ± 0.0421	0.372 ± 0.0024	0.459 ± 0.006
7	3M ESPE Filtek Z550 control	0.887 ± 0.0034	0.175 ± 0.003	0.211 ± 0.004
8	3M ESPE Filtek Z550 office bleach	3.058 ± 0.074	0.487 ± 0.0085	0.739 ± 0.0055
9	3M ESPE Filtek Z550 home bleach	1.356 ± 0.027	0.232 ± 0.0047	0.293 ± 0.0055
10	3M ESPE Valux Plus control	0.764 ± 0.0054	0.138 ± 0.0043	0.165 ± 0.0042
11	3M ESPE Valux Plus office bleach	1.63 ± 0.0354	0.311 ± 0.0024	0.37 ± 0.0453
12	3M ESPE Valux Plus home bleach	2.044 ± 0.0627	0.22 ± 0.0041	0.301 ± 0.0053

**Table 4 medicina-59-01067-t004:** Composite overall comparison and multiple group comparisons.

Composite	Overall*p* *	Group Comparisons
Control—Home Bleach	Control—Office Bleach	Home—Office Bleach
*p*	Trend	*p*	Trend	*p*	Trend
GC G-aenial anterior	<0.0005	<0.0005	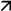	<0.0005	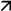	<0.0005	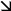
GC Gradia direct anterior	<0.0005	0.037	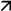	<0.0005	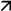	<0.0005	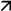
3M ESPE Filtek Z550	<0.0005	<0.0005	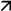	<0.0005	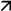	<0.0005	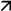
3M ESPE Valux Plus	<0.0005	<0.0005	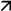	<0.0005	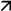	<0.0005	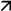

* One-way ANOVA.

**Table 5 medicina-59-01067-t005:** Mean roughness variation for each composite and bleaching type.

Bleaching Type	Composite	Mean ± SD	*p* *
Home	GC G-aenial anterior	0.374 ± 0.0048	<0.0005
	GC Gradia direct anterior	0.007 ± 0.0024	
	3M ESPE Filtek Z550	0.057 ± 0.0047	
	3M ESPE Valux Plus	0.082 ± 0.0041	
Office	GC G-aenial anterior	0.316 ± 0.0038	<0.0005
	GC Gradia direct anterior	0.550 ± 0.0039	
	3M ESPE Filtek Z550	0.311 ± 0.0084	
	3M ESPE Valux Plus	0.337 ± 0.0024	

* One-way ANOVA.

## Data Availability

The authors declare that the data from this research are available from the corresponding authors upon reasonable request.

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
