# Peer review of "Effects of Dental Bleaching Agents on the Surface Roughness of Dental Restoration Materials"

_medicina, 2023, doi:10.3390/medicina59061067_

Round 1

Reviewer 1 Report (Previous Reviewer 3)

Dear Authors of Medicine-2412833

 Effects of Dental Bleaching Agents on the Surface Roughness of Dental Restoration Materials

Here is presented an interesting article that seeks to determine if whitening agents are capable of producing rough surfaces in different resinous restoration materials, the methodology in my opinion is adequate, and results of interest to the clinician are presented.

However, there are several points that I would like to share with the authors with the utmost respect for their work:

1.- In the introduction, points that are important for the work are discussed, but I suggest giving relevance to the importance of having and maintaining polished surfaces in dental restorations instead of talking about dental caries, for example.

2.- The purpose at the end of the introduction is somewhat confusing since it states: "...two different types of composite materials". lines 102 and 103, in the sense that they are nanohybrid and micro-hybrid the statement is correct, however when reading the study, is evident that you used 4 composites (3 micro-hybrids, and one nanohybrid). I suggest changing this fragment for four composites or even putting their commercial names

3. In the discussion: in my opinion, the authors should mention how their findings are similar or how they differ from the articles presented, without the need to give a summary of the article to which reference is made, that important part is missing. Likewise, the authors should emphasize the importance of maintaining a polished restoration.

Solving these 3 aspects, his paper will be of great interest to the clinician.

Author Response

Response to Reviewer #1

27 May 2023

Dear Reviewer

It is an immense pleasure to submit our revised manuscript entitled: “Effects of Dental Bleaching Agents on the Surface Roughness of Dental Restoration Materials”.

We are grateful for the attention and effort spent in reviewing our work, and valuable comments made by the respectful editor and reviewers.

We sincerely hope that the revised manuscript is now suitable for publication in the Journal Medicina.

The authors ensured that the manuscript meets Medicina style requirements and have made changes to the manuscript according to the suggestions from the editor and the reviewers.

Find answers to your comments below.

1.- In the introduction, points that are important for the work are discussed, but I suggest giving relevance to the importance of having and maintaining polished surfaces in dental restorations instead of talking about dental caries, for example.

Dear reviewer, thank you for appreciating our work and for the advice offered. From the introduction I removed as you suggested the paragraphs related to dental caries and added the importance and implications of the clinical procedure of finishing and polishing (lines 58-65) I also made the appropriate changes in the bibliography.

Thank you very much for the advice.

2.- The purpose at the end of the introduction is somewhat confusing since it states: "...two different types of composite materials". lines 102 and 103, in the sense that they are nanohybrid and micro-hybrid the statement is correct, however when reading the study, is evident that you used 4 composites (3 micro-hybrids, and one nanohybrid). I suggest changing this fragment for four composites or even putting their commercial names

I made the changes in the introduction according to your advice (lines 100-102).

Thank you very much for the advice.

  1. In the discussion: in my opinion, the authors should mention how their findings are similar or how they differ from the articles presented, without the need to give a summary of the article to which reference is made, that important part is missing. Likewise, the authors should emphasize the importance of maintaining a polished restoration.

I followed your advice and made the changes during the discussions.

Thank you for the advice provided.

The results of the current study suggest that the roughness increased at the surfaces of all samples subjected to the bleaching protocols, in complete accordance with specialty literature [46,47].

Other studies have reported increases in the roughness of the surfaces of the resinous materials used, but also the appearance of cracks, pores and even scratches. The samples were analyzed using the electron microscope after undergoing the bleaching protocol using carbamide peroxide. In the present study, an increase in the surface roughness of the composites was observed following the application of 16% carbamide peroxide as part of the "home bleach" protocol [35,36,37].”

“Recent studies have been carried out on both traditional composites and bulk-type composites (with high viscosity as well as with low viscosity). These composites have undergone bleaching protocols with both 40% hydrogen peroxide and 10% carbamide peroxide. Following the testing carried out, it was concluded that the Ra (roughness) of the surfaces was higher after the bleaching procedures. Bulk-type composites with high viscosity recorded a higher roughness than conventional nanofilled and low viscosity ones. The increase of Ra’s following the application of 40% hydrogen peroxide was also observed in the present study as part of the "office bleach" protocol [53,54,55,56].”

“In 2008, Dogan et al. conducted a similar study, determining the roughness of some composite materials. Following testing, the researchers showed that the roughness of the surfaces of the bleached samples decreased, contrary to the results obtained in this study, where the roughness increased after applying the bleaching protocols [68].”

I also emphasized the importance of clinical finishing and polishing procedures on dental restorations with composite materials. (lines 415-421 and 436-439)

Thank you for the advice provided.

“Roughness on the surface of composites can also influence the color of dental restoration. Due to the increase in adhesion and retention of chromogenic food debris, the susceptibility of staining of the respective composites also increases, the gloss being subsequently affected. Other important elements that can influence the roughness of composites is represented by the dexterity of the dentist, the technique and system used during finishing and polishing plus the patients’ diet and hygiene, all these factors influence the roughness of the composites [44,45].”

“To combat this increase in roughness, some authors recommend polishing the dental obturations restored with composite materials after undergoing the bleaching protocol. A recent study showed that bacterial adhesion in gyomers is lower compared to other types of composite materials although roughness values were similar [51].”

We hope that the changes and explanations given will be to your liking!

Reviewer 2 Report (New Reviewer)

Manuscript title: Effects of Dental Bleaching Agents on the Surface Roughness of Dental Restoration Materials.

In this original research, authors have tried to compare the surface roughness of composite with different bleaching agents. However, there is research in the literature comparing the surface roughness of different composites with bleaching material. Hence, this research lacks novelty and originality. Authors should add more on the significance of the study.

Abstract: Well-written and within the word limits. Keywords: Add more keywords for readers to understand the study.

Introduction: A very basic introduction is written. For example, Line 45-47 does not make any sense, so many references are not required here. This introduction lacks flow in reading. It is hard to understand what authors want to convey. Rewrite the introduction and support the need for the study.

Material and method: Explain. There is no requirement of explaining the material and machinery utilized for experimentation in depth. Try to reduce it. Again, this section lacks flow in reading improve it.

Result: This section has been elaborated too much. Check the requirements for tables and figures.

Discussion: Null hypothesis mentioned is not clear. This section does not explain the importance of the findings of the study and the discussion does not support the result of the current study. Improve the discussion and write it clearly as it is difficult for readers to understand.

Overall, this manuscript is not written properly and is confusing for readers. The authors should improve all the IMRaD sections.

Needs moderate editing

Author Response

Response to Reviewer #2

27 May 2023

Dear Reviewer

It is an immense pleasure to submit our revised manuscript entitled: “Effects of Dental Bleaching Agents on the Surface Roughness of Dental Restoration Materials”.

We are grateful for the attention and effort spent in reviewing our work, and valuable comments made by the respectful editor and reviewers.

We sincerely hope that the revised manuscript is now suitable for publication in the Journal Medicina.

The authors ensured that the manuscript meets Medicina style requirements and have made changes to the manuscript according to the suggestions from the editor and the reviewers.

Find answers to your comments below.

  1. In this original research, authors have tried to compare the surface roughness of composite with different bleaching agents. However, there is research in the literature comparing the surface roughness of different composites with bleaching material. Hence, this research lacks novelty and originality. Authors should add more on the significance of the study.

Dear reviewer, thank you for the suggested comments, however, we would like to clarify certain aspects for the comments made.

- The present study was carried out on 4 different dental composites, while in the specialized literature the studies were focused on 2, maximum 3, types of composites.

- Regarding the realization of the bleaching protocols, we used hydrogen peroxide of 40% concentration (a higher concentration but accepted by the manufacturers) according to the "office bleach" protocol and carbamide peroxide of 16% concentration (a higher concentration but accepted by the producers) according to the "home bleach" protocol.

- In the specialized literature, I found only studies that used only one of the two protocols or lower concentrations. (Bibliography no. 16, 22, 31, 34, 36)

- Regarding the determination of surface roughness, most studies focus on the determination of a single parameter (Ra). In the present study, we determined three roughness parameters (Ra, Rz, Rq), thus being able to reach more accurate results and conclusions.

- Moreover, our study presents graphs of the roughness profiles, thus increasing the solidity of the study results. We consider that all these aspects give our study originality and novelty. Thank you very much for the advice!

  1. Abstract: Well-written and within the word limits. Keywords: Add more keywords for readers to understand the study.

We added new keywords as you suggested. Thank you very much for the advice! (row 41)

  1. Introduction: A very basic introduction is written. For example, Line 45-47 does not make any sense, so many references are not required here. This introduction lacks flow in reading. It is hard to understand what authors want to convey. Rewrite the introduction and support the need for the study.

Thank you for your comment.

I removed the paragraphs as you suggested, I improved the introduction by rearranging it to become more italic at the same time.

Thank you for the advice provided!

I also added the importance and implications of the clinical finishing and polishing process.

  1. Material and method: Explain. There is no requirement of explaining the material and machinery utilized for experimentation in depth. Try to reduce it. Again, this section lacks flow in reading improve it.

Thank you very much for the advice provided!

I took your advice into account and removed the details about the equipment used and improved the reading flow of the "Materials and methods" section.

  1. Result: This section has been elaborated too much. Check the requirements for tables and figures.

Thank you for your comment.

In this section we took into account the indications of the previous reviewers, that's why it turned out so complex and complete.

Thank you for your attention, I checked all the tables and figures in this study.

  1. Discussion: Null hypothesis mentioned is not clear. This section does not explain the importance of the findings of the study and the discussion does not support the result of the current study. Improve the discussion and write it clearly as it is difficult for readers to understand.

Thank you for your comment.

I made the changes suggested by you regarding the null hypothesis and the significance of the study. Thank you very much for the advice.

Discussion:

The null hypothesis was that the use of bleaching chemicals had no impact on the surface roughness of dental restorative composite materials. The null hypothesis is rejected by the findings of the preceding investigation. These observed changes in the surface roughness of the composites have clinical significance.

The study's novelty stems not only from the fact that a greater number of surface roughness parameters of the latest generation composites were determined, but also from the fact that bleaching materials were applied at a higher concentration, making comparison with previously published works difficult. This study informs clinicians about the level of risk associated with sustaining a composite restoration with increased surface roughness and the potential for plaque retention.

The clinical significance of the study: the findings imply that the alteration of the surface roughness of composites following the application of some whitening treatments necessitates the repolishing of the restorations or the placement of fresh composite fillings.

I followed your advice and made the changes during the discussions. Thank you for the advice.

  1. Overall, this manuscript is not written properly and is confusing for readers. The authors should improve all the IMRaD sections.

Thank you for your comment.

We took your suggestions into account and improved each section, giving the reader a better perspective on the quality of our work.

We hope that the changes and explanations given will be to your liking!

Round 2

Reviewer 2 Report (New Reviewer)

All the necessary corrections have been done

Good

This manuscript is a resubmission of an earlier submission. The following is a list of the peer review reports and author responses from that submission.

Round 1

Reviewer 1 Report

Especially the material method part of the related manuscript contains very serious methodological errors and non-compliance with the instructions for use of the relevant materials. Therefore, I think that the article should be rejected. However, the major and minor errors that have been revised are listed below as a guide to the authors for their future articles.

Abstract:

·      The first two sentences (lines 23 and 25) established for the purpose of the study should be combined. These sentences should be formally rewritten in passive language.

·      A brief description of the statistical methods applied should be written.

·      The p value should be mentioned in the results section.

·      A conclusion sentence should be written about the results. The current sentence is not suitable for this.

Introduction:

·      There is no sentence containing the purpose of the study. The null hypothesis is not established.

·      The difference of the study from other studies and which "niche" point in the literature is not specified.

·      There is a confusion of terminology throughout the manuscript. Composite resin or composite materials? It is not written in one language.

·      Passive language should be used instead of "we" in this part and all parts of the manuscript.

Materials and methods:

·      In Table 1, the explanations of the abbreviations of monomers such as BIS-GMA, UDMA should be written below the table. In the composition of Valux plus, the format should be written like the others, and the percentage should be indicated as in the others.

·      Was a standard pressure applied on the glass plate between lines 97-99? Is a weight placed on the glass plate for this? Or was this tried to be achieved with finger pressure?

·      It was stated that a single layer of composite resin was applied between lines 100-101. However, it is not recommended to apply these composites thicker than 2 millimeters. In this way, the polymerization is not fully realized, and the desired mechanical and physical properties cannot be expected from the related composite. This is one of the biggest methodological errors of the study.

·      Between lines 106 and 107, it was written that the composites were exposed to light for 20 seconds. However, these composites do not comply with the instructions for use. For example, 10 seconds of light is sufficient for G-aenial Anterior composites. Why was light applied to all composites (contrary to manufacturer's instructions) for 20 seconds? Is there any literature support on this?

·      It is stated that the super snap polishing system is used on lines 110-112. At how many RPMs was this polishing system used? Was the thickest polishing disc used at the same RPM? If so, did the thickest polishing disc cause any thinning of the composite samples? How was this thinning standardized? Were sample thicknesses re-measured?

·      How many seconds did it take to wash with the sterile saline solution mentioned in lines 117-118? Was an injector used for washing? How was it decided how much to wash?

·      Figure 1 is very blurry. It should be replaced with a clearer and higher resolution photo.

·      Descriptive sentences between lines 130 and 137 should be placed in an appropriate place in the introduction. These paragraphs are not suitable for the material and method part.

Results:

·      There are too many figures between figure 3 and figure 26. And these figures repeat each other. Figures that are similar should be combined and unnecessary ones removed from the manuscript. It is not understood.

Discussion:

·      At the beginning of this section, it should be written that the null hypothesis established is accepted or rejected.

·      Limitations of the study should be stated.

·      In this section, the authors gave too much and unnecessary details about other studies rather than discussing the findings of the study. This section is of no interest to the reader. It is too long and cannot be understood because it is out of context of the study.

Author Response

Dear Reviewer,

Thank you for your valuable suggestions.

We change the article according to your suggestions, as following:

Abstract:

  • The first two sentences (lines 23 and 25) established for the purpose of the study should be combined. These sentences should be formally rewritten in passive language.

This study aimed to evaluate the surface roughness evolution of several finished and polished composites when bleaching materials are applied. The research was conducted on four microhybrid or nanofilled composites that are used in dental restaurations.

  • A brief description of the statistical methods applied should be written.

The changes made regarding the statistical method are now described in the abstract section and also in the material and methods section.

  • The p value should be mentioned in the results section.

The p value is now reported in the results section.

  • A conclusion sentence should be written about the results. The current sentence is not suitable for this.

I made the necessary change

Introduction:

  • There is no sentence containing the purpose of the study.

The purpose of this study was to assess the surface roughness evolution of several finished and polished composites when bleaching materials were used. I put it in the text.

The null hypothesis is not established.

The null hypothesis of this study is that there is no significant difference in how tooth whitening materials modify dental restorative materials. I put it in the text.

  • The difference of the study from other studies and which "niche" point in the literature is not specified.

This study is significant because it will add to the literature on the mechanical changes undergone by composite materials under the action of tooth whitening substances and will provide clinicians with evidence-based recommendations. Ultimately, this can lead to improved clinical outcomes and a higher success rate for teeth whitening treatment.

  • There is a confusion of terminology throughout the manuscript. Composite resin or composite materials? It is not written in one language.

We eliminated the term ‘Composite resin’.

  • Passive language should be used instead of "we" in this part and all parts of the manuscript.

We made the adequate changes.

Materials and methods:

  • In Table 1, the explanations of the abbreviations of monomers such as BIS-GMA, UDMA should be written below the table. In the composition of Valux plus, the format should be written like the others, and the percentage should be indicated as in the others.

I have made the required changes to the abbreviations. In the case of Valux plus, the information presented in the tables is taken from the manufacturer's prospectus.

  • Was a standard pressure applied on the glass plate between lines 97-99? Is a weight placed on the glass plate for this? Or was this tried to be achieved with finger pressure?

The samples were made using a stainless-steel device, which produced composite discs that have 10 mm in diameter and 2.5 mm thickness. To make the samples, the composite material was applied in the device. The samples were prepared by compressing two glass plates to ensure uniformity and to eliminate air bubbles.

  • It was stated that a single layer of composite resin was applied between lines 100-101. However, it is not recommended to apply these composites thicker than 2 millimeters. In this way, the polymerization is not fully realized, and the desired mechanical and physical properties cannot be expected from the related composite. This is one of the biggest methodological errors of the study.

Between lines 106 and 107, it was written that the composites were exposed to light for 20 seconds. However, these composites do not comply with the instructions for use. For example, 10 seconds of light is sufficient for G-aenial Anterior composites. Why was light applied to all composites (contrary to manufacturer's instructions) for 20 seconds? Is there any literature support on this?

We used samples of 2.5 mm color A2, photopolymerized for 20 seconds with E. Woodpecker wireless lamp with a light intensity of 1000 mW/cm2, conforming exactly to the manufacturer's instructions.

We are attaching the manufacturer's instructions that we found.

  • It is stated that the super snap polishing system is used on lines 110-112. At how many RPMs was this polishing system used? Was the thickest polishing disc used at the same RPM? If so, did the thickest polishing disc cause any thinning of the composite samples? How was this thinning standardized? Were sample thicknesses re-measured?

Finishing was done at conventional RPMs with the four discs of different roughness as per the manufacturer's instructions. It was performed in order to avoid eventual surface roughness errors of the composite sample.

  • How many seconds did it take to wash with the sterile saline solution mentioned in lines 117-118? Was an injector used for washing? How was it decided how much to wash?

The samples were washed until all the bleaching agent was removed. A syringe of 10ml with saline solution of NaCl 0.9% was used.

  • Figure 1 is very blurry. It should be replaced with a clearer and higher resolution photo.

We made the appropriate change.

  • Descriptive sentences between lines 130 and 137 should be placed in an appropriate place in the introduction. These paragraphs are not suitable for the material and method part.

We made the appropriate change.

Results:

  • There are too many figures between figure 3 and figure 26. And these figures repeat each other. Figures that are similar should be combined and unnecessary ones removed from the manuscript. It is not understood.

I took into account your recommendation, and based on the values in the respective tables, I made graphs.

Moreover, in order to be clear and consistent with the conclusions, I have removed unnecessary figures.

Discussion:

  • At the beginning of this section, it should be written that the null hypothesis established is accepted or rejected.

We made the appropriate change.

  • Limitations of the study should be stated.

We made the appropriate change.

  • In this section, the authors gave too much and unnecessary details about other studies rather than discussing the findings of the study. This section is of no interest to the reader. It is too long and cannot be understood because it is out of context of the study.

I also reviewed the discussions.

Reviewer 2 Report

In this study authors have tried to find the influence of two bleaching protocols on surface roughness of various composite materials. But there is no mention about the rationale and not even the aim in the introduction of the manuscript. Material and methods section  also very vague with respect to the number of samples used and the methodology used. Results were not analysed to find out the influence of different bleaching protocols on each material as well as between the materials. The discussion is also very vaguely written not specific to your results

Author Response

Dear Reviewer,

Thank you for your valuable suggestions.

We change the article according to your suggestions, as following:

In this study authors have tried to find the influence of two bleaching protocols on surface roughness of various composite materials. But there is no mention about the rationale and not even the aim in the introduction of the manuscript.

We stated the purpose of the study at the end of the Abstract and Introduction section.

Material and methods section  also very vague with respect to the number of samples used and the methodology used.

We improved this section.

Results were not analysed to find out the influence of different bleaching protocols on each material as well as between the materials.

I have taken your recommendation into account. I have completed the results. I detailed the statistical study.

The discussion is also very vaguely written not specific to your results

We improved this section.

Please see the attachments below for the new article version!

Kind regards,

Prof. Dr. Mihaela Tuculina

Reviewer 3 Report

Effects of Dental Bleaching Agents on the Surface Roughness of Dental Restoration Materials

An interesting study is presented with the purpose of determining the roughness of composites after the use of whitening in the office and at home through evaluation techniques in accordance with the study that is proposed.

Several questions arise from reading your article.

Introduction

1. the purpose is only read in the abstract

2. the purpose should go at the end of the introduction and not as a statement with references

Materials and Methods

1. Regarding the methodology, how the sample was calculated

2. What references could you place that support that with a single sample of each composite you can reach the conclusions that you propose?

3. Is it a pilot study? Even when it is a confirmatory study, a calculation of the sample should be made to have a set that allows evaluating the behavior of the composites through a statistical treatment.

Conclusions

Can you reach these conclusions with a sample of each?

My intention is to make your work completely solid and your conclusions strong so that they can be cited without any problem. I think that by increasing the number of samples and reaching similar results your article could be accepted.

Author Response

Dear Reviewer,

Thank you for your valuable suggestions.

We change the article according to your suggestions, as following:

An interesting study is presented with the purpose of determining the roughness of composites after the use of whitening in the office and at home through evaluation techniques in accordance with the study that is proposed.

Several questions arise from reading your article.

Introduction

  1. the purpose is only read in the abstract
  2. the purpose should go at the end of the introduction and not as a statement with references

We took note of your suggestions and made the appropriate changes.

Materials and Methods

  1. Regarding the methodology, how the sample was calculated
  2. What references could you place that support that with a single sample of each composite you can reach the conclusions that you propose?

There were more than one sample used, thus we updated this section accordingly.

  1. Is it a pilot study? Even when it is a confirmatory study, a calculation of the sample should be made to have a set that allows evaluating the behavior of the composites through a statistical treatment.

This is not a pilot study. We further modified and clarified the statistical aspects of the current study.

Conclusions

Can you reach these conclusions with a sample of each?

We updated the text with respect to the actual number of samples used in the study.

Please see the attachments below for the new article version!

Kind regards,

Prof. Dr. Mihaela Tuculina

Round 2

Reviewer 1 Report

It is seen that minor errors and typos in the article have been corrected. The authors should be commended for this. However, I believe that the study should be rejected again, as errors arising from the design and methodology of the study cannot be corrected and some of my questions were answered partially rounded.

Reviewer 2 Report

The revised manuscript has more clarity on the rationale of the study, methodology and the results.